



# PAttern REtrieval or deNegation Testing Scheme (PARENTS) v.1.0 – Identifying the degree of presence of given patterns in spatial time series

**Benjamin Müller[1,2], Matthias Bernhardt[1], and Karsten Schulz[1]**

[1]Institute for Hydrology and Water Management (HyWa), University of Natural Resources and Life Sciences, Vienna, Austria

[2]Department of Geography, Ludwig-Maximilians-Universität, Munich, Germany

*Correspondence to*: Benjamin Müller (b.mueller@iggf.geo.uni-muenchen.de)

## 10  Abstract

The large number of spatially distributed earth observation products, i.e. time series of surface emissions and reflectances at different wavelengths with increasing spatial resolution, contribute to the derivation of surface characteristics, e.g. vegetation or soil parameters in the environmental sciences. These derivatives usually build upon complex algorithms consisting

of atmospheric corrections and process descriptions.

The testing scheme presented here seeks a different approach to identifying these surface characteristics that control the generation of such observation time series. Spatially distributed patterns of these characteristics of different persistence usually dominate parts of a time series because of their very specific reaction to and interaction with environmental influences. We

test these characteristics' patterns for their existence in a rotated vector space of elementary patterns derived from a principal component analysis of an observational time series. With the result of this test we can then make valid assumptions, e.g. with regard to the importance of the surface properties for the emittance or reflectance, or their spatial uncertainties.

We demonstrate the functionality of this rather simple test algorithm for a synthetic and fully

traceable example, and an application in a medium hydrological catchment for a time series of thermal satellite data. Possible future applications for this scheme are the prioritization and improvement of model input, data assimilation, or the evaluation and validation of model output.



## 1   Introduction


With the current advancements in satellite, aircraft and drone remote sensing as well as in geophysical measurement techniques, spectral reflectance and emissivity information of the land surface are available from an increasing number of platforms at different spatial and temporal resolutions (Weng, 2017). By applying complex atmospheric corrections,

reflectance and emissivity models, land surface reflectivity (visible, near-infrared, radar) and emissivity (thermal, microwave) data are translated into relevant environmental variables, such as surface temperature ($T_s$), soil water content (WC), leaf area index (LAI), snow cover (SC) or evapotranspiration (ET) (Bastiaanssen et al., 1997; Brenner et al., 2017; Dorigio et al., 2017; Dong, 2018; Horn and Schulz, 2010), among many others.

The spatial and temporal patterns of these variables are strongly controlled by land surface characteristics such as land surface energy balance, relief, or the availability of soil moisture and therefore the general climate, vegetation and soil properties. Identifying the dependencies between land surface characteristics and the directly remotely sensed land surface information is important to improve our environmental process understanding and to advance the

predictive capabilities when remote sensing data are used to predict land surface characteristics or processes (see e.g. Awange et al., 2006; Matgen et al., 2006; Müller et al., 2014, 2016; among many others).

In this context, it is often necessary to identify the specific control or impact that individual patterns of land surface characteristics (e.g. vegetation, or soil texture; in the following called

"parental pattern", PP) have on the full time series of patterns of system states (e.g. $T_s$ or WC images; in the following called TS), or on just individual elements/images of the TS. A common tool to analyze these controls is to calculate simple 1-to-1 comparison scores based on an individual PP and single TS members. Possible scores include correlation, root mean square error, or normalized distribution similarity tests (e.g. Kolmogorov-Smirnov test;

Corder and Foreman, 2014). However, when applying simple comparison scores, PPs are only compared to single members of the TS and it is not possible to quantify the overall effect of any PP on the complete TS.

The complete TS of e.g. surface temperature, snow or water content pattern represents the dynamic behavior of the observed land surface system and can be regarded as an *n*-

dimensional vector space of possible states, where *n* is the number of elements (remotely sensed images) within the TS. While each element of TS will contain some important



information about the land surface system under investigation, a certain degree of redundancy in the information content of each TS member has to be expected due to the dominance of certain PPs.

Principal component analysis (PCA) is a statistical procedure to analyze the correlation structure between several observations (members of TS). A PCA converts a set of (possibly highly) correlated observations into a set of linearly uncorrelated variables (principal components, PCs) using an orthogonal transformation (Richards and Jia, 2006). The first PC is in the direction of the largest part of the data variance, and each subsequent PC has the

highest variance possible under the constraint that it is orthogonal to the preceding components. The resulting set of PCs therefore constitute an uncorrelated orthogonal basis, whereby the individual PCs are ordered by "the portion of variance of the original they explain". In this way, it is possible to extract the most dominant components (patterns) of the observed system, which is represented by the TS.

The control or impact of any PP on the TS (e.g. the impact of spatially distributed soil texture pattern on a time series of surface temperature) can be assessed by quantifying the agreement (or the degree of deviation) of the PP with the individual PCs. However, a possible direct agreement between a PP and any of the PCs is largely disrupted by any noise in the measurements and/or signal-processing step. A PP will only be exactly found or reconstructed

(or identical to any PC) for completely noise-free data, and when the PPs are independent of each other with distinct weights (deduced from linear regression without noise; compare Howard, 2010). However, these circumstances are hardly found under real conditions.

As the objective is to identify the most relevant and controlling PPs for the TS of interest, we circumvent the aforementioned difficulties by developing a new method that is based on the

following step: First, a PCA is performed on the TS and a set of $n$ orthogonal PCs is derived. Second, we additionally optimize a $n$-dimensional rotation of the system of PCs with regard to the net angle between the selected PP and the primary PC. In this way, we receive for each PP the one vector that is best correlated to it, and the angles to all PCs can be extracted.

We will explain the general idea as well as the mathematical formulation of this procedure in

more detail in section 2. A synthetic test case in section 3.1 will demonstrate the principle functionality of the method, while in section 3.2, it is applied to a real case problem, where a time series of thermal remote sensing images is analyzed to extract their dominant landscape





controls. Further application areas and an outlook are provided in section 4. Section 5 finally
will provide some guidance to access the archive of available code.


## 2 Method description

### 2.1 Problem and method definition

Let us assume, we have a complex system that is based on a number of $m$ PPs (e.g.
vegetation, soil texture) as shown in Fig. 1 (upper panels; here $m=2$: $B_1$, $B_2$). The PPs may
"generate" a series (TS) of patterns (e.g. patterns of surface temperature $T_s$, that are a result of
some complex interaction within the water, radiation and energy balance of a landscape).
Such a series of possible patterns $I_1$, $I_2$, …, $I_n$ is shown in Fig. 1 (lower panels, $n=6$), whereby
in this synthetic case, the patterns are generated as random linear combinations of the PPs $B_1$
and $B_2$ and some additive gaussian noise.

The question now is which of the PPs is contributing to what degree to the observed pattern
$I_1$-$I_n$, or in other words, to what degree can the PPs be reconstructed or generated by the TS $I_1$-
$I_n$. In order to answer this question, we use the complete set $I_1$-$I_n$ of available "observations"
that arrange an $n$D vector space based on this set. The angles between these vectors are
normally non-orthogonal, i.e. members of $I_1$-$I_n$ are interrelated and dependent on each other,
and do not permit any conclusions about their importance. To avoid these dependencies and
to provide a ranking system, we orthogonalize the $n$D vector space by applying a PCA and
extracting the PCs. The now orthogonal vector space permits i) quantifying a degree of
importance by defining the smallest angle between a reconstructed or generated PP and PC1,
the principle component with the largest share of the overall variability, ii) a finer
determination of transitions between different states due to the same angular distances
between the PCs, and iii) a reduction of the $n$D vector space to an $(n-k)$D vector space, by
declaring the $k$ PCs with the smallest share as noise or "not important enough". Thus, we
improve above declaring a PP by i) quantifying its importance with ii) a high resolution
gradient and iii) reducing calculation effort.

### 2.1.1 Principal component analysis

The herein used principal component analysis (PCA; a complete mathematical description can
be found at Richards and Jia, 2006, chapter 6.1), or similarly empirical orthogonal functions





(EOFs, e.g. Denbo and Allen, 1984; Hamlington et al., 2011; Lorenz, 1956) are widely used within the field of analyzing big data (Fan and Bifet, 2013; Feldman et al., 2013). Hence,

these mathematically similar methods are often used in the fields of climate sciences (meteorology: Preisendorfer and Mobley, 1988; geography: Demšar et al., 2013), especially with remote sensing data (Pohl and Van Genderen, 1998).

In general, PCA is not applicable for 2D members of the nD vector space. To analyze 2D patterns, these patterns have to be flattened to 1D vector. Thus, a linearization procedure has

to be defined for decomposition and (re)production of a 2D version of the resulting PCs. Additionally, these flattened patterns are normalized (Fisher z-transformed) to allow equal weighting of all patterns. Then, a covariance matrix and, subsequently, eigenvectors are calculated. The eigenvectors, sorted by size, define the described partial variance, principal components' rotation matrix, and, hence, loadings for calculation of the orthogonal PC vector

space.

For non-dependent and unscaled PPs, as well as non-noisy PPs and TS data, PCA can directly be used for near-exact reproduction of PPs. However, real measurement data contains noise, so that dependencies between PPs and nonlinear scalings are to be expected. Therefore, the following procedure is added.

**2.1.2 Rotation**

Rotation is needed to find patterns between the axes of the PCA (=PCs) in case noise prevents direct reproduction. The rotation of a $n$D vector space can be described by multiplication of the vector space with a $(n \times n)$-sized rotation matrix $R$ with the properties $R^T = R^{-1}$ and

$det R = 1$ (Howard and Rorres, 2010). A rotation of all $n$ orthogonal vectors of the $n$D vector

space is conformal, hence, results in a new $n$D orthogonal vector space (Howard and Rorres, 2010). Thus, original patterns can be found similarly to finding them in the original orthogonal $n$D vector space.

However, only the rotation of the first PC is important as orthogonal rotations can be added to rotate other PCs as well. Thus, the absolute angles of $R$ are interpreted as a measure of

deviation from each PC and, therefore, the deviation from each more or less dominant pattern within the original $n$D vector space.


If a rotation is unknown and to be determined by a pre- and an exact post-rotation vector, the rotation matrix can directly be calculated by

$$x' = \begin{bmatrix} x_1{'} \\ \vdots \\ x_n{'} \end{bmatrix} = \begin{bmatrix} r_{1,1} & \cdots & r_{1,n} \\ \vdots & \ddots & \vdots \\ r_{n,1} & \cdots & r_{n,n} \end{bmatrix} \begin{bmatrix} x_1 \\ \vdots \\ x_n \end{bmatrix} = Rx \text{ with } \frac{x'x}{|x|^2} = R.$$

Although, $R$ can be assessed directly, we need to consider noise in both observations (PCs) and PPs that are to be examined. Therefore, an optimization of $r_{1,1} \ldots r_{n,n} \equiv R$ is necessary. For simplification, we assume $r_{i,j} = r_{j,i}$, which is compliant with a proper definition of rotation and halves the number of parameters to be optimized.

### 2.1.3 Optimization and scores

Optimization of the rotation matrix in this approach is accomplished by applying an extended version of the Dynamically Dimensioned Search (DDS) algorithm, described by Tolson and Shoemaker (2007). This robust and effective algorithm is commonly used for optimization of numerous parameters for complex hydrological models (Wallner et al., 2012). This algorithm is chosen due to its single perturbation parameter, hence reduced tuning effort, and highly 165 efficient search for good, but not optimal, solutions, hence low computational cost. For this application, we extend DDS by a loop of decreasing perturbation parameter to refine the optimum search. This approach is inspired by the simulated annealing schedule (Kirkpatrick et al., 1983).

The optimization itself is based on a predefined error function. Here, we use a maximization 170 of the (absolute) Pearson's correlation score between a potential PP and the rotated PC1. This score is used as (linear) scaling and orientation (=sign) do not need to be specified for the calculated error. Thus, linear scaling and orientation adjustments are applied in the final stage for visualization of rotated patterns, so values are within the original ranges (see Figures in Section 3.1).

We emphasize that the chosen optimization algorithm and error function may not be optimal for the following examples, but are chosen for their simplicity and traceability. As Wolpert and Macready (1997) state, there is not a single optimization algorithm that is fitting to a large or, much less, full range of different problems and/or data sets.





The described procedure in sections 2.1.1-2.1.3 is further called PARENTS (**PA**ttern
**RE**trieval or de**N**egation **T**esting **S**cheme) as it is capable of both reproducing actual and
eliminating faulty PPs.

## 2.2 Data processing scheme and output

The PARENTS algorithm in this approach is specifically used for analyzing time series of
spatial patterns from remote sensing. Though, it can easily be adjusted for non-geocoded data
sets.

The data are processed in the following way: First, the TS data set is assumed to be
preprocessed as a 3D data stack (dimensions: latitude, longitude, time). If necessary, the data
has to be resampled/georeferenced. Also, the considered PP is resampled/georeferenced to the
same domain and resolution as the 3D data stack. Then, PCA is performed on the TS data set.
Further, the PCs and the PP(s) are rearranged as a matrix, i.e. losing their spatial setup while
being able to reproduce the original spatiality. This matrix is then multiplied by a random
rotation matrix, which then is optimized with the extended DDS algorithm to reach a high
correlation between the rotated and rescaled and optionally inverted PC1 and the PP(s). The
final output is the rotated PCs sorted in a decreasing order, the rotation matrix, the correlation
score, and net angles of the PP(s) to the different PCs for each PP.

There is an additional option to reduce the number of PCs used for the rotation optimization.
Reducing the PCs significantly reduces processing time, while, in the case of specifically
large partial variances in the first few PCs, of little loss.

## 3 Test cases

In the following, we present two test cases: one synthetic data set, which already was
introduced in sect. 2.1, with a controlled setup of paternal patterns and noise, and one
representing a typical application in remote sensing.

### 3.1 Synthetic data set

#### 3.1.1 Input

The synthetic data set is generated by different linear combinations of the two orthogonal 2D
patterns $B_1$ and $B_2$ (Fig. 1, upper panels) with additional white noise. Pattern $B_1$ contributes


with a normally randomized weight of $\mu_{w1}=10$ ($\sigma_{w1}=\ln(10)$), $B_2$ with a weight of $\mu_{w2}=5$ ($\sigma_{w2}=\ln(5)$), and noise with a weight of $\mu_{w3}=2$ ($\sigma_{w3}=\ln(2)$) to generate the data set patterns.

These resulting patterns are then linearly scaled to the same value range as the original patterns $I_1$-$I_6$ (see Fig. 1, lower panels).

The data set is tested for the original PPs $B_1$ and $B_2$, as well as a third pattern N (checkerboard, Fig. 3, upper right panel) that is orthogonal to both PPs. N is used as a denegation dummy in this example.

**3.1.2 Results**

First, we analyze the benefits of the PARENTS algorithm when compared to only applying the PCA without rotation. For this, we gradually increased the noise component (weight of noise, Fig. 2) when generating arbitrary sets of $I_1$-$I_6$ (see Fig. 1) while fixing the weights of $B_1$ and $B_2$ for the different set elements. Fig. 2 (middle panel) illustrates the reconstructed pattern

of the higher weighted $B_1$ (Fig. 1, upper left) when only applying a simple PCA and extracting the first principle component PC1.

It is easily seen that with increasing noise ("weight of noise", w.o.n) the reproduction of $B_1$ is blurred and the correlation coefficient between $B_1$ and PC1 is decreasing from values ~0.9 down to ~0.8 (see Fig. 2, upper panel). The PARENTS algorithm, including an optimized

rotation of the PC system, however is able to reproduce $B_1$ almost independent of the noise with correlation coefficients very close to 1.0 (see regression line in Fig. 2, upper panel), indicating a strong improvement in terms of robustness against and overall quality of the reconstruction. Though, the reproduction of $B_1$ is of lower quality per definition of the PCA, when the influence is quite close to other patterns (see Fig. 2, w.o.n=10).

The reproduction of all PPs from the more randomly constructed TS (as shown in Fig. 1) with the PARENTS algorithm using an extensive optimization (1.000.000 steps) is shown in Fig. 3. The PPs $B_1$ and $B_2$ can be almost exactly reproduced with an absolute correlation score close to 1. Though, pattern $B_1$ appears inverted in the final result. This is due to the error function being defined as absolute correlation score. The reproduction of a checkerboard

pattern N - that is not used for production of the data set and is orthogonal to $B_1$ and $B_2$ - cannot be achieved; the absolute correlation score is as low as 0.05.

In summary, it can be stated that the PARENTS algorithm has successfully been tested with a synthetic setting, where the PPs as well as the time series of TS were perfectly well known.



This example was limited in the number of TS members, as well as of PPs, but was perfectly
suited to demonstrate the functioning of the methodology, especially under different noise
conditions. The second example will address a more realistic case study, where a time series
of thermal remote sensing images over a hydrological catchment in Luxembourg is
investigated to analyze the dominant controls on surface temperature.

### 3.2   Typical application with real data set - the Attert catchment and thermal
remote sensing data

The realistic case is based on the Attert catchment study within the German DFG research
project CAOS ("catchments as organised systems"; CAOS, 2019). The 288 km² catchment is
located in mid-western Luxembourg and southeastern Belgium (see Fig. 4).  Initial studies on
the usage of PCs for grouping the catchment into hydrological functional units and deriving
soil texture characteristics were performed in Müller et al. (2014, 2016).

### 3.2.1   Input

The utilized basic TS data set consists of 28 thermal infrared remote sensing images from the
multispectral imaging system ASTER (advanced spaceborne thermal emission and reflection
radiometer) on board the TERRA satellite from between January 2001 and June 2012 (see
Fig. 5). The satellite orbits on a near circular, sun-synchronous path with a repeat cycle of 4-
16 days and was launched December 1999. The ASTER instrument's three sensors cover the
wavelengths of VNIR (visible-near infrared: 0.52-0.86 µm), SWIR (shortwave infrared: 1.6-
2.43 µm), and TIR (thermal infrared: 8.125-11.65 µm) with 4, 6 and 5 bands, respectively
(Fujisada, 1995). The Level 1A TIR product, band 13 (10.25-10.95 µm), with an original
spatial resolution of 90 m is used, due to its most complete preservation of ground thermal
patterns (compare Elder and Strong, 1953).

The potential PPs that are assumed here are those landscape characteristics that are likely to
influence land surface temperature and therefore the radiation, energy and water balance of
the catchment. A resolution adjusted digital elevation model (DEM; ACT, 2013) and a
derived hill shade (see Neon22, 2014) will have an impact on the incoming shortwave
radiation, but also on the distribution of soil moisture via water transport processes
redistributing incoming precipitation. A geological map with dominant rock formations (SGL,
2003) might be a reasonable proxy for soil formation processes and soil texture distribution
that water holding capacity and other hydraulic and thermal properties. CORINE land cover





(EEA, 1995) data give information about vegetation patterns in the catchment, which have an
      important impact on the energy and water balance via albedo, water uptake and
      evapotranspiration. We also added a random uniformly distributed pattern as a denegation
      dummy.

      The hill shade PP consists of the average values for the 28 actual hill shades at flyover (11:30

am) based on slope and aspect of the DEM. The geological and the land cover data is
      originally of nominal characteristic. In many applications, these are used for class based
      parameter estimation. Thus, the nominal data is replaced by a numeric value for the
      PARENTS algorithm. The rock formation classes are used to appoint thermal inertia of the
      underlying bedrock, whereas land cover is translated to LAI. The set values can be found in

tables 1 and 2.

      In the next section, the PARENTS algorithm will be applied to investigate whether and to
      what extend the assumed PPs can be reconstructed by the time series of thermal images, or, in
      other word to what extent is each PP controlling the dynamics of the TS.

### 3.2.1 Results

Figure 6 illustrates the results applying the PARENTS algorithm to the full TS data set of 28
      thermal images and the 5 potential PPs height, hill shade, thermal inertia, LAI and random
      noise for the Attert catchment. The algorithm ran for 20.000 optimization steps for each PP.
      While the upper panel of Fig. 6 shows the "original" patterns of the PPs, the reconstructed
      PPs are given in the middle panel showing their respective deviation from the original PPs.

The lowest panel shows a smooth per-pixel scatter plot, including a 1:1 line, and the
      regression line with correlation coefficient, for each PP.

      The highest correlation between PPs and the rotated PC1 of the TS (0.81) can be found for the
      "height" (above sea level) information (Fig.6, lower panel); the valleys as well as the
      relatively high altitude in the northwest of the catchment can be well recognized within the

rotated PC. Results for the "height above sea level" illustrate the importance of adiabatic
      temperature gradients on the temperature signal in hilly or mountainous terrains and are
      therefore expected to affect all TS images.

      We can find similar structures within the "hill shade" and its rotated counterpart. "Hill shade"
      is relevant for the TS patterns by highlighting areas where generally more radiation is

available and heating up the surface. As this hill shade pattern is based on temporal averages





over all images, we expect a rather low quality of reproduction. Here, the PARENTS algorithm is finding especially highly shaded areas (≥0.75) with higher deviations in the rotated PC1 and, thus, only low correlation (r=0.4).

The pattern of "thermal inertia" of the underlying bedrock is expected to have only a marginal
influence on TS. The underlying bedrock is buffering heat fluxes within the overlying soil with temporal offset. "Thermal inertia" shows a similar spatial gradient compared to "height" with low values in alluvial valleys and high values in the schists heights. The correlation between both original PPs is relatively high (r=0.78). As this PP is not a continuous pattern, correlation with the rotated PC is lower (r=0.55) when compared to the height pattern, as
expected.

Vegetation and, thus LAI, is a very strong control on thermal patterns of surface temperature, due to its capability of regulating stomata resistance and, hence, the amount of transpiration, thereby cooling the leaf surface. The rotated PC for the land cover class based LAI still shows a high correlation (0.63) to this originally non-continuous pattern.

Finally, and as expected, the uniform noise cannot be reproduced (r=0.05), again indicating the good performance of the PARENTS methodology.

In addition to the correlation coefficients between PP and the rotated PC1 in Fig. 6, Table 3 provides an additional subset of rotation angles between the PPs and PC2 and PC3. The data show that the LAI patterns resembles PC1 at best, while PC2 shows to be closer to thermal
inertia. For PC3, the angles show only low direct resemblance to the tested PPs. These findings fully support the results from Müller et al. (2014; section 3.3, Fig. 10) where PC1 is connected to CORINE and PC2 to geological maps.

The effect of reduction of the number of PCs, here to a maximum of 5 (compare Müller et al., 2014), can be found in Fig. 7. The reduction of the number of PCs limits the optimization
problem to less parameters (5x5 instead of 28x28 => ~factor of 30) while potential information is lost, that could be found in the angular transitions between the higher PCs. This brings benefits within the computational effort for the cost of reproduction accuracy. Still, reducing the system at this point has less important information lost than reducing the length of the original TS.

The correlation scores for height and thermal inertia drop the most. This leads to the assumption that height and rock information is distributed among PCs of lower importance.



The correlation score for LAI is practically the same, as the pattern is very close to PC1 (compare Müller et al., 2014).

Beyond the reproduction of potential patterns, we also explored the potential of reproducing a
single image of the TS (rather than of PP) that was either included or excluded from the observation time series. This analysis was performed by considering the full set of PCs and compare it to results when only the first 5 PCs where were taken. We chose three different images for this test: a summer image (15 Jul 2008), a winter image (15 Feb 2003), and one especially clouded image (26 Sep 2003). Tables 4 shows the correlation results for these
experiments. As summer situations are well represented in the overall data set (compare Fig. 5), the correlation is high for both included and excluded, as well as the reduced number of PCs. Only minor structural features cannot be reproduced. Winter situations are not very well presented in the full TS, so excluding one image results in a similar effect on its final correlation as the reduction of the number of PCs. The specific winter patterns are of low
importance for the overall variations. Though, excluding one winter image result in a lower correlation than reducing the number of PCs. Clouded images usually show an untypical pattern for the investigated area and, hence, show the largest drop in correlation results for the exclusion experiments.

We see, that the PARENTS algorithm is capable of reproducing potential PPs of a thermal
remote sensing TS to a reasonable quality. At the same time, noise cannot be reproduced, what underlines the meaningfulness of the internal system structure of the algorithm and its results. Further, we showed the degree of loss of information, when limiting the number of PCs. This is useful, when only PPs of high importance need to be found or if exclusion experiments are needed first. Last, we examined the influence of images with very specific or
unusual compared to common patterns in the TS. This can also be used to reduce the number of dimensions and thus, computational effort or descriptive complexity.

## 4    Conclusion

In this paper, we present the PARENTS algorithm to identify the impact of any PP on an
observed TS. We demonstrated the performance of the method using a synthetic data set and showed that including an optimized rotation will reproduce a pattern better than a simple PCA, when more than one important pattern is present. The PARENTS algorithm was also successfully tested in real case study when identifying the importance of different PPs on time


series of thermal images. A number of experiments analyzed the performance of the method
when constraining the number of PCs in the rotation/optimization process.

The PARENTS algorithm shows a very good performance in reproducing patterns in the
synthetic setup. The original patterns are retrieved with high correlation scores for arbitrary
influence of noise, compared to the simple PCA, and non-existing patterns cannot be
reproduced (Fig. 2-3). Based on these results, the algorithm is tested within a realistic data set
of 28 thermal remote sensing images and four different reasonable potential PPs and one
uniform noise pattern. The main find is, that the random pattern is irreproducible (r=0.05),
while the reasonable PPs can be found with different, though relatively high correlation scores
(0.4-0.81). Patterns that are physically directly connected to the thermal surface signal, such
as "LAI" or "height", have higher scores than geological or shading patterns (Fig. 6). While
correlation scores do not explain causality between PP and TS, the rotation angles within the
vector space claim the patterns' importances for the data set.

The reduction of the computational effort by reducing the number of PCs shows a decrease in
information compared to the full range of PCs (compare Fig. 6 and 7). Thus, this modified
approach can be used to perform either preliminary examinations of a TS for denegation of
specific patterns or weighing the importance of the PPs for the TS more specifically.

We also show that the quality of reproduction is directly connected to a pattern's
representation in the underlying data set (see table 4). The TS itself consists of mainly
summer and spring, and less winter images. At the same time, partial clouds are not recurring
at the same positions and extents. Still, such described "outliers" can be (re)produced with
some lower quality and, hence, lower certainty.

We therefore expect the PARENTS algorithm to identify PPs successfully throughout
different scales, as long as the underlying data set is covering a large variety of possible
naturally occurring patterns. The algorithm, as presented, is a functional tool for
understanding the contribution of surface characteristics in processes and their weights within
an remotely observed system.

A large number of further applications of the algorithm seems possible, e.g. in the field of
remote sensing of the environment:

- The use within assimilation schemes in (earth system) modeling. A number of
  different remote sensing data can be used as basis for the PARENTS algorithm to





search a modeled pattern (present or future) of a distributed model. This tested pattern then can be adjusted to the closest available reasonable pattern from the data basis.

- The use within spatial interpolation of unknown values. It is possible to use the PARENTS algorithm for the search of partially existing patterns or point measurements and fill the void, from e.g. clouds, with the pattern of the rotated PC

based on the available pixels.

- The refinement of observed broad classes. The LAI patterns, for example, are well represented within the rotated PC. Still, adjustments of the scoring function to e.g. preserve average LAI values for each class is needed.

- The use within solving downscaling issues. A coarse pattern to be downscaled to a

finer resolution can similarly be executed as the refinement of class based patterns.

The algorithm can be further adjusted by including non-linear scaling of the rotated PC, though this adds additional optimization effort, or changing the scoring function to also exceed the tested value range (extrapolation). These proposed applications are scope for future research and will be addressed in the near future.


## Acknowledgements

We thank the German Research Foundation (DFG) and the Austrian Science Fund (FWF) for funding this research through the CAOS (Catchments as Organised Systems) Research Unit (FOR 1598; SCHU1271/5-1; I2142-N29). We also want to thank the LPDAAC (Land

Processes Distributed Active Archive Center) for providing free ASTER data as well as the editor and anonymous referees for their contributions to improve this article.

## Code/Data availability

Processed data is available from the authors on request. The authors plan to publish the code in a public github repository or as a Python or R package in near future.

**Author contribution**

Benjamin Müller designed the algorithm and the computational framework and analyzed the data. All authors contributed to the final version of the manuscript. Karsten Schulz supervised the project.

## Competing interests

The authors declare that they have no conflict of interest.



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





**Table 1: Numeric values (thermal inertia) for spatial classes within the geological map and their origin. These values from Robertson (1988) are assigned to the referenced map (SGL, 2003).**

| spatial class | assigned value | | composite value(s) |
|---|---|---|---|
| alluvials and deposits | 42 | $[10^{-3} \text{ cal cm}^{-2} \, ^\circ\text{C s}^{0.5}]$ | 42 (~clay soil) |
| sandstone | 54 | $[10^{-3} \text{ cal cm}^{-2} \, ^\circ\text{C s}^{0.5}]$ | 54 |
| marls | 45 | $[10^{-3} \text{ cal cm}^{-2} \, ^\circ\text{C s}^{0.5}]$ | 45 (~limestone) |
| schists | 63 | $[10^{-3} \text{ cal cm}^{-2} \, ^\circ\text{C s}^{0.5}]$ | 63 (~serpentine) |



**Table 2: Numeric values (LAI) for spatial classes within the CORINE land cover and their origin. These values from (1) Sun and Schulz (2017) and (2) Knote et al. (2009) are assigned to the referenced map (EEA, 1995). Assigned values are estimated modes for the used image dates.**

| spatial class | assigned value | | composite value(s) |
|---|---|---|---|
| agricultural-natural mix | 2.60 | [m² m⁻²] | 0.98 – 3.17 (2) |
| arable | 2.26 | [m² m⁻²] | ~0.4 –5.0 (1); 0.68 – 2.78 (2) |
| artificial | 0.87 | [m² m⁻²] | 0.44 – 2.15 (2, continuous and discontinuous urban) |
| complex cultivation | 2.15 | [m² m⁻²] | ~0.5 – 4.5 (1) |
| coniferous forest | 2.91 | [m² m⁻²] | 1.67 – 3.32 (2) |
| deciduous forest | 2.89 | [m² m⁻²] | ~1.0 – 6.0 (1); 1.21 – 3.45 (2) |
| mineral extraction | 2.20 | [m² m⁻²] | 0.81 – 2.24 (2) |
| mixed forest | 3.02 | [m² m⁻²] | 1.57 – 3.50 (2) |
| pastures | 1.95 | [m² m⁻²] | ~0.2 – 5.5 (1); 1.12 – 3.06 (2) |






**Table 3: Angles resulting after rotating the first 5 PCs to the rotated PC. Higher deviations from 90° have to be interpreted as patterns closer to the compared PC. Values origin from the same run as Fig. 6.**

| angle to PC | height a.s.l. | hill shade | thermal inertia | LAI | uniform noise |
|---|---|---|---|---|---|
| PC1 | 98.6° | 85.1° | 96.4° | **60.6°** | 90.1° |
| PC2 | 89.7° | 91.7° | **102.7°** | 89.7° | 94.0° |
| PC3 | 90.0° | 91.2° | 89.8° | 92.5° | 89.9° |





**Table 4: Correlation result for one thermal image from the TS and the rotation for different exclusion setups. Patterns were included (in) or not (out), and all PCs were used or just the first 5. Highest correlations are found for the inclusion of all images while lowest correlations are found for the exclusion of the specific image while reducing the number of PCs to the first 5. The decline is larger when images are less represented by the remaining TS.**


| Image | summer (15 Jul 2008) | | winter (15 Feb 2003) | | cloudy (26 Sept 2003) | |
|---|---|---|---|---|---|---|
| Setup | in | out | in | out | in | out |
| Correlation for 28/27 images in TS | 0.998 | 0.944 | 0.998 | 0.757 | 0.995 | 0.703 |
| Correlation for first 5 PCs from 28/27 images | 0.878 | 0.862 | 0.796 | 0.743 | 0.705 | 0.681 |



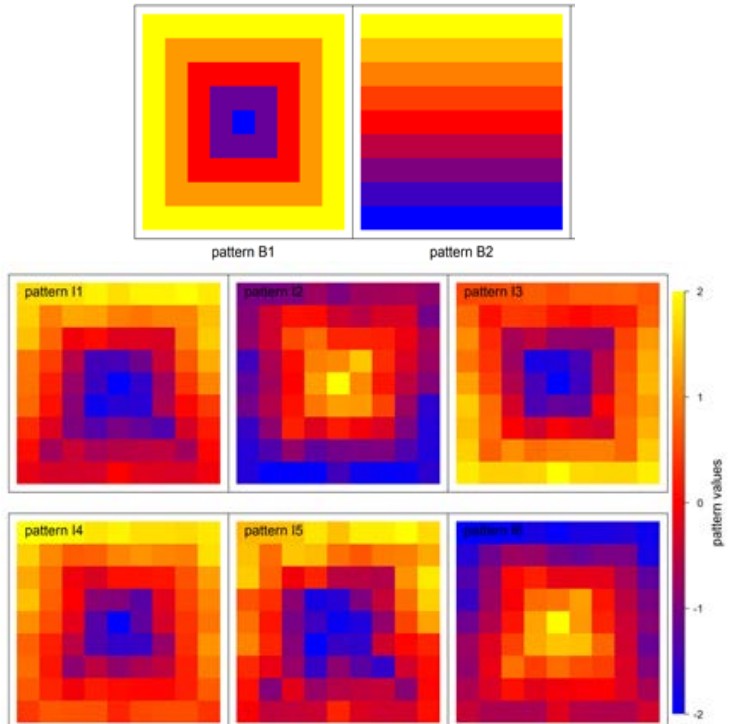

**Figure 1: The basic orthogonal 2D patterns B₁ and B₂ for use as PPs in the synthetic data set (upper panels) and the synthetic data set I₁-I₆ of 2D patterns generated from PPs B₁, B₂, and gaussian noise (lower panels). Pattern B₁ contributes with a normally randomized weight of μ_{w1}=10 (σ_{w1}=ln(10)), B₂ with a weight of μ_{w2}=5 (σ_{w2}=ln(5)), and noise with a weight of μ_{w3}=2 (σ_{w3}=ln(2)) to generate the data set patterns. For better comparison, these resulting patterns are linearly scaled to the same value range as the original patterns.**



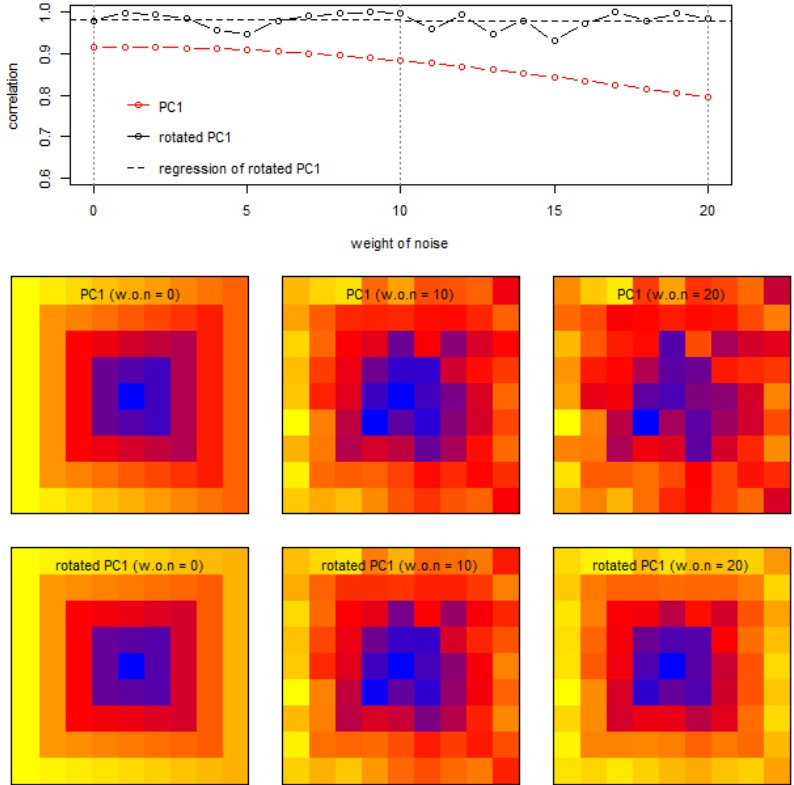

**Figure 2: Assessment of the influence of noise for the difference between optimized rotation and PCA only. For generating the analyzed data sets, weights for the patterns $B_1$ and $B_2$ were fixed with little variation ($w_{1,1-6}$ = 10+[0:0.1:0.5], $w_{2,1-6}$ = 5-[0:0.1:0.5]) while a noise pattern was fixed with increasing weight for the different sets. The upper panel shows the correlation scores after 1.000.000 optimization steps for PC1 (red circles) and rotated PC1 (black circles) and $B_1$ for the different weights of the noise. Additionally, the regression line (dotted) of the correlations of rotated PC1 is added to emphasize the near constant scores of almost 1.0. The lower panels show exemplary results of PC1 (top) and rotated PC1 (bottom) for the weights 0, 10 and 20 (compare $B_1$ in Figure 1, upper panel).**





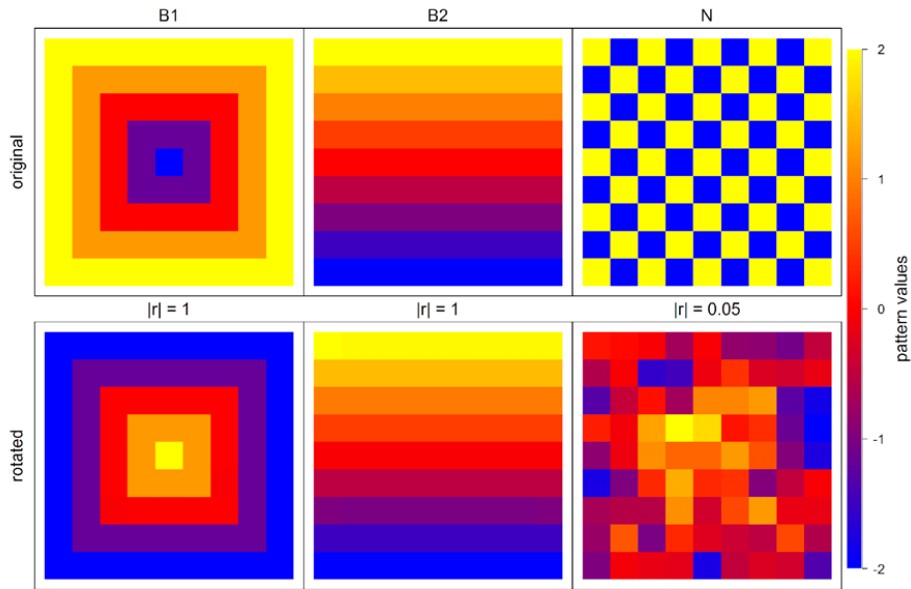

**Figure 3: An exemplary reproduction (bottom) of the potential PPs (upper panel, compare Fig. 1) from the data set (Fig. 1, lower panel) with an extensive optimization routine. The checkerboard pattern (N) cannot be reproduced. Correlation results (|r|) are noted between rotation results and original patterns.**

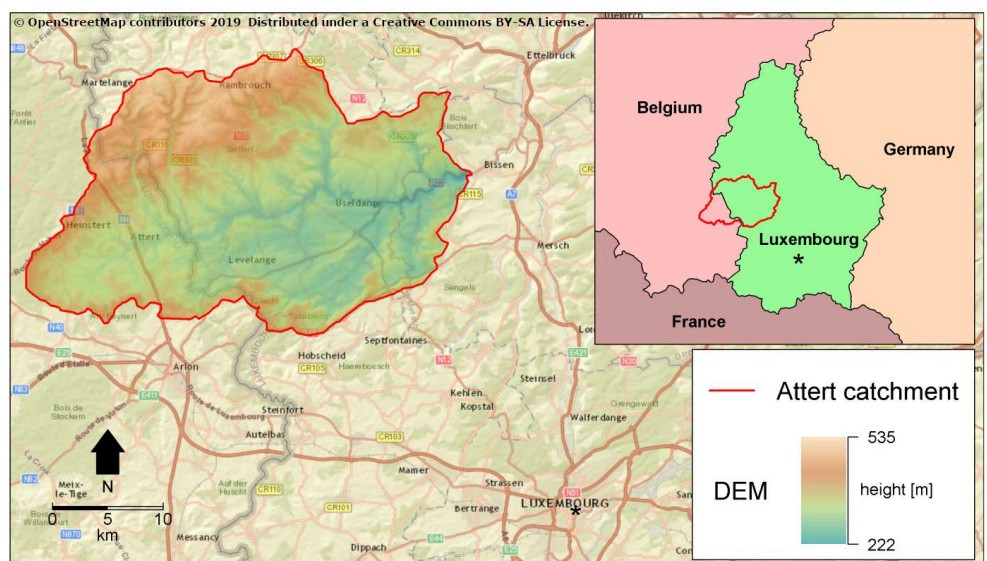

**Figure 4: The location of the Attert catchment with additional information on elevation. Catchment boundaries are defined by the location of the gauge Bissen, Luxembourg. For more information about catchment characteristics, the reader is referred to Müller et al. (2014, 2016). Basic map from © OpenStreetMap (Distributed under a Creative Commons BY-SA License, https://www.openstreetmap.org/copyright).**

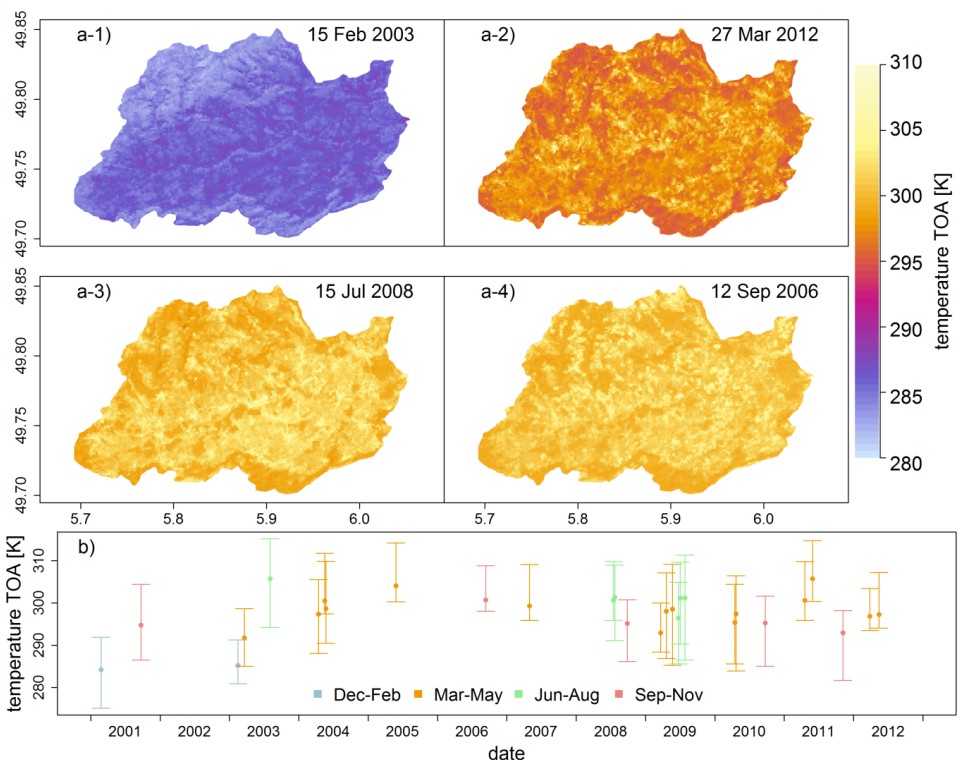

Figure 5: a) Examples of the seasonal TS data set patterns (winter 1, spring 2, summer 3, and autumn 4). b) The timing and ranges of data set patterns throughout the TS.

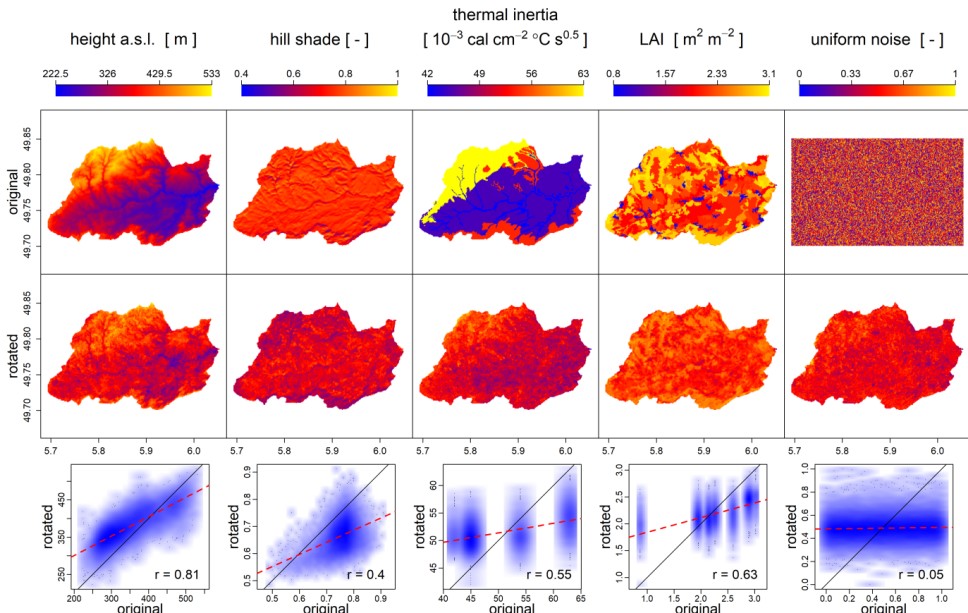

**Figure 6: Result from the PARENTS algorithm for the full TS of 28 images and the potential PPs height**
**(above sea level), hill shade, thermal inertia, LAI and a uniform noise pattern. First row shows original**
**potential PPs, second row shows the rotated PC, and the third row shows the smoothed scatter plots of the**
**patterns above with 1:1 line (black) and linear model (red, dashed) for better comparison.**



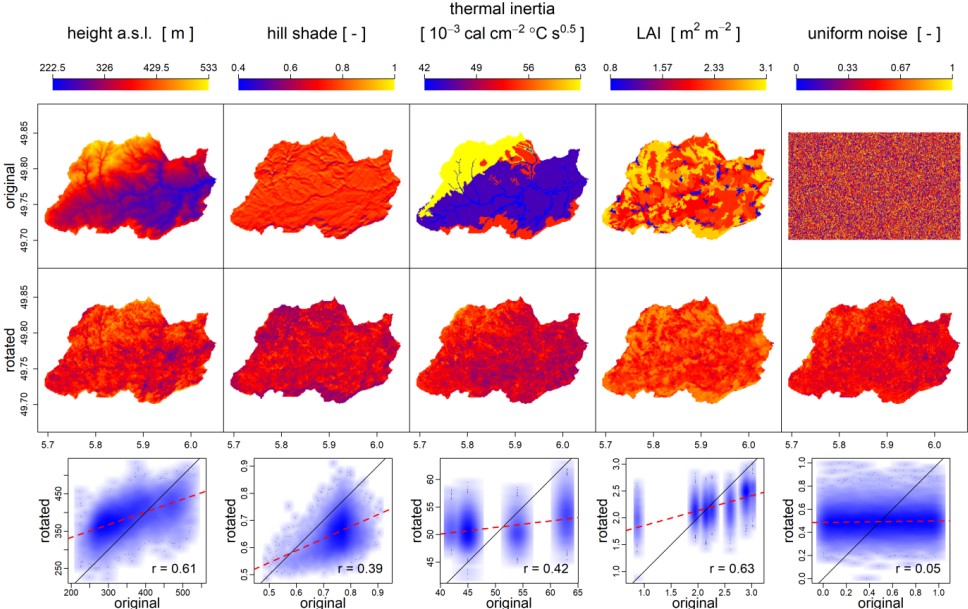

**Figure 7: Result from the PARENTS algorithm for the full TS of 28 images and the potential PPs height (above sea level), hill shade, thermal inertia, LAI and a uniform noise pattern. The number of PCs is limited to the first 5 most affecting (compare Müller et al., 2014). Illustration is set up according to Fig. 6.**