# Peer review of "PAttern REtrieval or deNegation Testing Scheme (PARENTS) v.1.0 – Identifying the degree of presence of given patterns in spatial time series"

_Hydrology and Earth System Sciences, 2019_

## Referee Comment (RC1) · Anonymous Referee #1 · 29 Jan 2020

General comments:

A relatively straightforward PCA method is used to retrieve biogeophysical variables from a time series of 28 ASTER images acquired from 2001 to 2012 covering a catchment. Title suggests that a software is presented ("Parents") but this software is not available now. This paper is a third of a series of studies using the same dataset over the same area. Which remotely sensed variable is used is unclear because several terms are used ("surface temperature", "land surface temperature", "temperature TOA"). Introduction does not permit understanding the novelty and the objectives of

this work with respect to previous studies by the same authors. Figure 4 was already published (Figure 1 in Mueller et al. 2014 and Figure 1 in Mueller et al. 2016). Figure 2 was already published (Figure 2 in Mueller et al. 2014 and Figure 4 in Mueller et al. 2016). Finally, I tend to think that this work is not technically correct. While previous works focused on rather static properties of the catchment such as land cover, topography, geology, soil texture, this study considers thermal inertia and LAI. Thermal inertia depends on soil moisture. Both soil moisture and LAI can vary rapidly and both present seasonal and interannual variability. I do not see how they can be monitored using data presenting such a poor sampling time. The authors would rather use geology and land cover as they did in their previous work (but then: what's new?). This paper cannot be published in the present form.

Recommendation: reject.

Particular comments:

- L. 37: Only surface soil moisture (SSM) can be observed from space. "WC" is too vague.

- L. 51: What does "TS" stand for? Time series ?

- P. 3: Authors should explain the added value of this paper with respect to previous studies published in HESS (2014 and 2016) by the same authors. Is there anything new in this study?

- L. 278-280 (and Tables 1 and 2): This is not correct. Moisture could influence thermal inertia (not only bedrock properties) and LAI has seasonal and interannual variability.

- L. 393 (data assimilation): not clear for understanding. Which kind of DA schemes?

- L. 535: what is the definition of a LAI "composite value"?

- L. 566 (1000000 optimization steps): sounds a lot.

- L. 585 (Figure 5): what does TOA stand for? Top Of the Atmosphere? In this case,

you should refrain calling this quantity "surface temperature".

- L. 590 (Figure 6): Thermal inertia and LAI are variables through time. What do these maps mean?

---

## Referee Comment (RC2) · Anonymous Referee #2 · 14 Feb 2020

This paper describes a testing "scheme [which] seeks a different approach to identifying these surface characteristics that control the generation of such observation time series". Right away, I am confused about the objective of the paper. The sentence above is a good example – what approach, what surface characteristics, and what observation time series are the authors' talking about?

Abstract and the first few sentences of the paper do not clearly state the objectives, or even a problem statement that the paper will address. The Introduction, the first paragraph makes a general statement, but does not point to a problem or issue that

the work attempts to answer or solve.

One very simple reason why this paper is difficult to understand is that the authors are using indirect language to describe their ideas, which obfuscates their meaning. An example of this is the use of a lot of acronyms very early on in the paper. The composition would be much stronger if the authors simplified the paper to remove extensive use of acronyms.

At some point in the Introduction, the authors start to talk about 'TS'. So, perhaps this is a focus of their work, but it is not clear. They also randomly talk about PCA, which may or may not be an interesting way to examine TS. But, again, it is not clear.

The paper has unconventional organization. I suggest following a more standard flow. For example, having two results sections is confusing. These could be combined to make it easier for the reader to follow. There is no discussion of the results, per sec. A discussion would be nice, and perhaps this could help the authors to organize their ideas from Introduction to Discussion and Conclusions.

I think if this paper were rewritten to have clear objectives, and cleaner /clearer flow of ideas and expression, it would be easier to follow the reasoning in the paper from concept to results/discussion.

My recommendation is to return this paper to the authors for major revision.

―――――――――――――――

---

## Author Comment (AC1) · 6 Mar 2020

Answer to Anonymous Referee #1 ():

Original comment in plain text, answers in italic.

General comments:

A relatively straightforward PCA method is used to retrieve biogeophysical variables from a time series of 28 ASTER images acquired from 2001 to 2012 covering a catchment. Title suggests that a software is presented ("Parents") but this software is not available now.

*We will be happy to adjust the title accordingly to avoid confusion:*

**Identifying the degree of presence of given near-surface characteristics in time series of remote sensing images – Introduction and description of the PAttern REtrieval or deNegation Testing Scheme (PARENTS)**

This paper is a third of a series of studies using the same dataset over the same area. Which remotely sensed variable is used is unclear because several terms are used ("surface temperature", "land surface temperature", "temperature TOA").

*We will homogenize the wording throughout the document with additional details on the differences and assumptions that were no yet repeated from the predecessor papers.#M3*

Introduction does not permit understanding the novelty and the objectives of this work with respect to previous studies by the same authors.

*Given that both reviewers indicated some difficulties in understanding the objectives and research questions of work, we decided to apply mayor revisions on the introduction and the abstract, to clearer describe the problem we address and to clarify how the methods evolved from Müller et al. 2014 and 2016 to this paper.*

Figure 4 was already published (Figure 1 in Mueller et al. 2014 and Figure 1 in Mueller et al. 2016). Figure 2 was already published (Figure 2 in Mueller et al. 2014 and Figure 4 in Mueller et al. 2016).

*We feel that both figures are essential in describing the catchment and the used data. If the suggestion is, to simply cite them without showing, we are happy to do so, but feel it would be much easier for the reader to have that information straight at hand.*

Finally, I tend to think that this work is not technically correct. While previous works focused on rather static properties of the catchment such as land cover, topography, geology, soil texture, this study considers thermal inertia and LAI. Thermal inertia depends on soil moisture.

*The reviewer is absolutely correct when pointing out that thermal inertia is dynamically dependent on soil moisture. The herein used thermal inertia is solely the bedrock part of the integral soil thermal inertia, and thus rather constant. We will emphasize this explicitly in a revised version of the manuscript.#M1*

Both soil moisture and LAI can vary rapidly and both present seasonal and interannual variability.

*LAI cannot be directly measured for a catchment of this size and the used data is a theoretical snapshot and mainly used as an example. We will emphasize that intention in a revised version.#M2*

I do not see how they can be monitored using data presenting such a poor sampling time.

*This is mainly the novelty of the approach. We do not attempt to monitor soil moisture or LAI directly, but we attempt to find related patterns in the transition angles between the PCs, which are based on limited sampling time. We will emphasize this approach in the revised introduction.*

The authors would rather use geology and land cover as they did in their previous work (but then: what's new?).

*For both characteristics, we need a numeric representation, which is stated in the paper. Thus, the need for a translation to exemplary thermal inertia and LAI patterns.*

This paper cannot be published in the present form.

Particular comments:

- L. 37: Only surface soil moisture (SSM) can be observed from space. "WC" is too vague.

*"WC" is the used term in the cited reference. Will be changed in the revised introduction.*

- L. 51: What does "TS" stand for? Time series?

*The abbreviation will be introduced in a clear manner after revision.*

- P. 3: Authors should explain the added value of this paper with respect to previous studies published in HESS (2014 and 2016) by the same authors. Is there anything new in this study?

*Will be part of the revision of the introduction.*

- L. 278-280 (and Tables 1 and 2): This is not correct. Moisture could influence thermal inertia (not only bedrock properties) and LAI has seasonal and interannual variability.

*See marks M1 and M2 in the general comment above.*

- L. 393 (data assimilation): not clear for understanding. Which kind of DA schemes?

*We will make clear how the scheme could be used in DA schemes. (In short, the idea is to use pattern nudging instead of value nudging, with patterns of modelled data being replaced by patterns that can be found in an observation time series with the presented scheme.)*

- L. 535: what is the definition of a LAI "composite value"?

*Term will be adjusted to "literature values".*

- L. 566 (1000000 optimization steps): sounds a lot.

[Figure]

*This is information solely for comparative measures (i.e. all optimization runs had the same number of iterations). Thus, 1 Mio steps is not compute time/cost intensive in nowadays machines and not unexpected for a set of 12 free parameters within a range of 0°-360°.*

- L. 585 (Figure 5): what does TOA stand for? Top Of the Atmosphere? In this case, you should refrain calling this quantity "surface temperature".

*See mark M3 in the general comment above.*

- L. 590 (Figure 6): Thermal inertia and LAI are variables through time. What do these maps mean?

*See marks M1 and M2 in the general comment above.*

**Thank you for your valuable input and constructive comments.**

––––––––––––––––––––––––––––

---

## Author Comment (AC2) · 6 Mar 2020

Answer to Anonymous Referee #2 ():

Original comment in plain text, answers in italic.

This paper describes a testing "scheme [which] seeks a different approach to identifying these surface characteristics that control the generation of such observation time series". Right away, I am confused about the objective of the paper. The sentence

above is a good example – what approach, what surface characteristics, and what observation time series are the authors' talking about? Abstract and the first few sentences of the paper do not clearly state the objectives, or even a problem statement that the paper will address. The Introduction, the first paragraph makes a general statement, but does not point to a problem or issue that the work attempts to answer or solve.

*The need for a much clearer presentation of the objective was raised by both reviewers. We therefore decided to apply mayor revisions on the introduction and the abstract, to make clear how the methods evolved from Müller et al. 2014 and 2016 to this paper, instead of the current writing approach.#M1*

One very simple reason why this paper is difficult to understand is that the authors are using indirect language to describe their ideas, which obfuscates their meaning.

*We will revise the text with respect to the use of direct formulations, where applicable.*

An example of this is the use of a lot of acronyms very early on in the paper. The composition would be much stronger if the authors simplified the paper to remove extensive use of acronyms.

*We will reduce the use of acronyms, especially if they are of limited recurrence.*

At some point in the Introduction, the authors start to talk about 'TS'. So, perhaps this is a focus of their work, but it is not clear. They also randomly talk about PCA, which may or may not be an interesting way to examine TS. But, again, it is not clear.

*See mark M1 in the comment above.*

The paper has unconventional organization. I suggest following a more standard flow. For example, having two results sections is confusing. These could be combined to

make it easier for the reader to follow.

*We discussed different structures throughout the writing process and found the current structure - separating the two case studies - to be the better solution. However, we will improve the guidance through the structure at the end of the introduction.*

There is no discussion of the results, per sec. A discussion would be nice, and perhaps this could help the authors to organize their ideas from Introduction to Discussion and Conclusions.

*The discussion in this paper was merged with the results to avoid too many subsections. This will be emphasized.*

I think if this paper were rewritten to have clear objectives, and cleaner /clearer flow of ideas and expression, it would be easier to follow the reasoning in the paper from concept to results/discussion. My recommendation is to return this paper to the authors for major revision

**Thank you for your valuable input and constructive comments.**